# Advanced Triboelectric Applications of Biomass-Derived Materials: A Comprehensive Review

**DOI:** 10.3390/ma17091964

**Published:** 2024-04-24

**Authors:** Chan Ho Park, Minsoo P. Kim

**Affiliations:** 1Department of Chemical and Biological Engineering, Gachon University, 1342 Seongnamdaero, Sujeong-gu, Seongnam-si 13120, Republic of Korea; 2Department of Chemical Engineering, Sunchon National University, Suncheon 57922, Republic of Korea

**Keywords:** biomass, triboelectric device, flexible, biodegradable

## Abstract

The utilization of triboelectric materials has gained considerable attention in recent years, offering a sustainable approach to energy harvesting and sensing technologies. Biomass-derived materials, owing to their abundance, renewability, and biocompatibility, offer promising avenues for enhancing the performance and versatility of triboelectric devices. This paper explores the synthesis and characterization of biomass-derived materials, their integration into triboelectric nanogenerators (TENGs), and their applications in energy harvesting, self-powered sensors, and environmental monitoring. This review presents an overview of the emerging field of advanced triboelectric applications that utilize the unique properties of biomass-derived materials. Additionally, it addresses the challenges and opportunities in employing biomass-derived materials for triboelectric applications, emphasizing the potential for sustainable and eco-friendly energy solutions.

## 1. Introduction

### 1.1. Introduction to Biomass-Derived Materials

Historically, biomass-derived materials have been pivotal in various applications [1,2,3,4] when natural resources were the primary materials for human needs due to their natural abundance and versatility. The evolution of these materials can be traced from rudimentary uses in construction and textiles to more sophisticated applications in energy and environmental technologies. Biomass materials, essentially derived from biological sources like plants, microorganisms, and agricultural residues, showcase a diverse array of chemical and physical properties [5]. These properties hinge on their origin and processing methods, presenting a wide spectrum ranging from cellulose-based fibers with high tensile strength to lignin-rich substances known for their robust thermal stability. These materials have gained renewed interest due to their sustainability, biodegradability, and abundant availability.

The definition of biomass-derived materials emphasizes their renewable and biodegradable qualities [6,7]. These materials, including cellulose, lignin, chitosan, and natural fibers, possess a complex composition of cellulose, hemicellulose, and lignin, which determines their mechanical, chemical, and thermal properties. Their versatility is evident in applications such as environmentally friendly energy-harvesting devices. For instance, cellulose nanofibers from wood pulp are used in triboelectric nanogenerators (TENGs) due to their light weight, high surface area, and piezoelectric properties, offering a sustainable alternative to artificially synthesized materials that lack eco-friendliness.

### 1.2. Triboelectric Devices: Definition and Mechanism

The triboelectric effect, a type of electrostatic induction, is the foundational principle behind triboelectric devices [8,9,10]. When two different materials come into contact and are then separated, they generate an electric charge due to the transfer of electrons. This phenomenon is central to the operation of TENGs. Typically, triboelectric materials are categorized based on their electron affinity, which defines their ability to either gain or lose electrons [11,12,13,14,15,16]. Common materials used in TENGs include metals, synthetic polymers, and now, increasingly, biomass-derived materials.

The output efficiency of triboelectric devices is fundamentally linked to the density of surface charges [17]. To maximize this efficiency, it is imperative to ensure not only the generation of surface charges through contact electrification but also their effective transfer via electrostatic induction. This underscores the importance of choosing the right materials for the triboelectric pairs and crafting the device structure with precision [9,12,16]. Reflecting on their operational fundamentals, four distinct models of triboelectric devices have been identified: the vertical contact-separation mode, the in-plane contact-sliding mode, the single-electrode mode, and the freestanding triboelectric-layer mode [18]. All these models utilize dielectric materials in their triboelectric layers.

In triboelectric devices that employ a dielectric-to-dielectric configuration (Figure 1a), two dielectric plates characterized by distinct thicknesses (*d*_1_ and *d*_2_) and relative dielectric constants (*ε_r_*_,1_ and *ε_r_*_,2_) are aligned in a face-to-face manner to form the triboelectric layers, with electrodes being applied to the outer surfaces of these dielectrics. The separation between these triboelectric layers is dynamically adjusted through the application of a periodic mechanical force. This interaction leads to the generation of opposite charges on the surfaces of the triboelectric layers that come into contact, maintaining an equivalent charge density (*σ*) due to the process of contact electrification. As the triboelectric layers start to move apart because of an increasing separation, a potential difference (*V*) emerges between the electrodes due to the transfer of positive and negative charges (+*Q*/−*Q*). In a similar vein, the conductor-to-dielectric mode, which excludes the first dielectric layer, utilizes a metal as both the primary triboelectric layer and the electrode. This configuration results in metal 1 harboring two types of charges: those (*S* × *σ*) generated triboelectrically and those (−*Q*) transferred between the electrodes, contributing to the aggregate charge within metal 1 (*Sσ* − *Q*). The output efficiency of these devices can be analyzed through the principles of electrodynamics (Figure 1b). Here, the concept of effective dielectric thickness (*d*_0_) comes into play, calculated as the aggregate thickness of the dielectrics (*d_i_*) divided by their cumulative relative permittivity (*ε_r,i_*). This calculation is crucial as it directly impacts the triboelectric performance through its influence on the surface charge density of the dielectric layers. This approach underscores the interplay between material properties and device architecture in optimizing the performance of triboelectric devices.

Enhancing the surface charge density of triboelectric materials has been a focal point for improving the efficiency of triboelectric devices [19,20,21,22]. Strategies such as the surface modification of these materials or incorporating materials with a higher dielectric constant have been proposed. By adjusting the surface morphology of triboelectric materials, adding charged ions, or integrating electron-withdrawing/donating groups into the polymer chains [23,24], the surface charge density can be increased. These modifications serve to either expand the surface area or enhance the triboelectric contrast between differing triboelectric materials. Furthermore, optimizing surface characteristics is not the sole method for enhancing performance; elevating the dielectric constant of materials also plays a critical role in increasing surface charge density. This is attributed to the augmented capacitance of the dielectric layer, a critical aspect for amplifying surface charge density in devices where dielectric-to-dielectric contact is employed (Figure 1c) [19,25]. Since the capacitance, which plays a key role in boosting the surface charge density, rises with an increase in the dielectric constant and/or a reduction in the dielectric layer’s thickness, the surface charge density benefits from a higher ratio of dielectric constant to layer thickness [19,23,24,26,27].

The integration of biomass-derived materials into triboelectric applications marks a significant step forward [28,29,30]. Their inherent properties, like biodegradability, flexibility, and natural triboelectric characteristics, make them ideal candidates for TENGs. For instance, cellulose-based materials have shown promising triboelectric performances, attributed to their high electron-donating capabilities and structural stability.

In exploring the synergy between biomass-derived materials and triboelectric devices, we present a comprehensive overview of this emerging field. The applications range from self-powered sensors, which capitalize on the biocompatibility and flexibility of these materials, to energy harvesting systems that leverage their natural abundance and renewability. The burgeoning interest in triboelectric materials for sustainable energy harvesting and sensing technologies marks a significant advancement in the field of material science and energy engineering, spotlighting their role in energy harvesting, self-powered sensors, and environmental monitoring. This review examines the challenges and opportunities in utilizing biomass-derived materials for triboelectric applications. Critical challenges include enhancing the durability and efficiency of these materials in TENGs, which is essential for their wider adoption. Conversely, significant opportunities are present in exploring the untapped potential of various biomass sources for conversion into high-performance triboelectric materials.

## 2. Biomass-Derived Triboelectric Devices

In recent years, the realm of energy harvesting has experienced a notable shift towards sustainable and environmentally friendly methodologies. Among the diverse strategies employed, triboelectric energy harvesting emerges as a standout due to its proficiency in transforming mechanical energy into electrical power. The utilization of biomass-derived materials within this field is especially promising, providing a means to transform abundant, inexpensive, and renewable resources into precious energy. This strategy not only addresses the challenges associated with waste management but also makes a vital contribution to the advancement of renewable energy technologies. This section delves into triboelectric energy harvesting devices crafted from biomass-derived materials, exploring both their fundamental operating mechanisms and recent advancements in their application.

### 2.1. Cellulose-Based Triboelectric Devices

Cellulose, the most abundant organic polymer on Earth, derived from plant cell walls, has emerged as a promising material in the field of triboelectric energy harvesting [28,31,32]. Its renewable nature, biodegradability, and excellent mechanical properties make cellulose an ideal candidate for developing sustainable energy harvesting devices. Triboelectric energy harvesting, which relies on the triboelectric effect and electrostatic induction, benefits significantly from cellulose-based materials due to their inherent electron-donating properties, which enhances the charge transfer process. This section investigates the application of cellulose and its derivatives in TENGs, focusing on the integration of cellulose either as a component of the triboelectric layers or as a substrate supporting other active triboelectric materials.

Nanocellulose or cellulose derivatives (Figure 2a) have been used as active triboelectric layers in TENGs [31,33]. Nanocellulose, which can be used in various forms, including cellulose nanofibers (CNF) and cellulose nanocrystals (CNC), with its high surface area and mechanical strength [34], offers an enhanced triboelectric performance [35]. Most researchers have investigated how surface modifications or chemical treatments enhance the triboelectric performance by increasing the surface charge density and improving the mechanical durability [33]. Nie et al. explored the impact of various terminal functional groups on the triboelectric performance of CNFs by introducing electron-donating and electron-withdrawing units [36] (Figure 2b). The chemical modifications employed included the attachment of 3-aminopropyltriethoxysilane (APTES), 3-mercaptopropyltriethoxysilane (MPTES), 3-cyanopropyltriethoxysilane (CPTES), triethoxy-1H,1H,2H,2H-tridecafluoro-n-octylsilane (PFOTES), N-Methylaminopropyltrimethoxysilane (NMAPS), and (N,N-dimethylaminopropyl)trimethoxysilane (NNMAPS) to CNFs. The study’s findings revealed that the effectiveness of these functional groups in donating electrons is ranked as follows: -NH_2_ > -SH > -CN > -CF_2_CF_3_. Additionally, the study noted that the basicity of ammonia molecules modified with different hydrocarbon groups—classified as primary, secondary, and tertiary amines—influences the triboelectric output, with the propensity to donate electrons arranged in the order of R_3_N > R_2_NH > RNH_2_. This observation underscores the principle that electron-donating functional groups increase the electron cloud density and its spatial occupation, whereas electron-withdrawing groups result in a more compact electron cloud [37]. Consequently, based on their electron-donating or -accepting characteristics, the chemically modified CNFs can serve as either positive or negative materials in triboelectric applications [36,38,39,40,41].

These findings underscore a crucial strategy for material engineering within triboelectric applications. By customizing the chemical composition and functionality of nanocellulose fibers, researchers can exert effective control over the electron dynamics at the interface. Such control facilitates the precise adjustment of the material’s triboelectric properties, which may optimize the TENG performance. Moving forward, it would be advantageous for subsequent research to investigate the long-term stability and environmental ramifications of these chemically modified nanocellulose materials in practical settings. Furthermore, the integration of these materials into composite structures could potentially unveil synergistic effects that enhance the triboelectric output beyond the capabilities of single modifications.

In addition to the chemical modification of cellulose derivatives, the utilization of cellulose composites, particularly those blended with high-dielectric-constant (high-*k*) materials, marks a notable advancement towards crafting high-performance and eco-friendly triboelectric materials [42,43]. Such composites significantly enhance the effective dielectric constant, thereby elevating the surface charge density, a crucial factor for the efficiency of TENGs, as mentioned in the Introduction Section (Figure 1). By merging cellulose with a variety of materials, these composites capitalize on the natural attributes of cellulose and the distinctive properties of the additives to boost both the efficiency and the longevity of TENGs. The formulation of cellulose composites for triboelectric purposes involves a deliberate process of integrating conductive nanoparticles [44,45], polymers [46,47], or other functional materials [48,49] into a cellulose matrix (Figure 3). This strategic selection of additives aims to augment the electrical conductivity, mechanical robustness, and surface texture of the composite. For example, the integration of metal nanoparticles or conductive polymers can markedly elevate the composite’s charge generation and transmission abilities [45]. Additionally, incorporating nanostructured materials can increase the surface area, thereby facilitating enhanced contact electrification [48]. The triboelectric output of cellulose composites is influenced by an array of factors, such as the nature and number of additives, the surface morphology, and the prevailing environmental conditions. Demonstrating promising outcomes in TENG applications, cellulose composites have been shown to achieve high efficiency in power generation, enduring mechanical stress, and adaptability to diverse environmental settings. This evolution in the development of cellulose composites underscores their potential to revolutionize the field of energy harvesting through TENGs, pointing towards a sustainable future with efficient, durable, and environmentally conscious triboelectric materials.

On the other hand, cellulose composites, recognized for their flexibility and biodegradability, have increasingly been employed as substrates to support triboelectric materials [50]. By merging cellulose with a variety of additives, including biopolymers [51] and conductive nanomaterials [52,53,54] (Figure 4), these composites are engineered to exhibit enhanced mechanical flexibility, superior electrical properties, and improved surface characteristics. The effectiveness of cellulose composites in triboelectric applications is largely influenced by their surface morphology and electrical conductivity. The integration of nanostructured materials or conductive polymers not only expands the surface area but also bolsters the electron-transfer capabilities of the composites. This, in turn, results in a heightened charge density and an uptick in energy conversion efficiency. Such enhancements render cellulose composites as ideal substrates for triboelectric layers, effectively broadening their application scope. Additionally, the biodegradability of cellulose, combined with its widespread availability and affordability, significantly contributes to the environmental appeal of these materials [48,55]. These advancements are in line with the escalating demand for environmentally friendly energy solutions, marking a significant step forward in the development of sustainable energy harvesting technologies. As the field advances, cellulose composites are poised to play a crucial role in the future of green energy technologies, particularly in the realms of wearable electronics and other innovative applications.

The integration of cellulose composites with high-k materials and various conductive additives represents a significant leap forward in the field of TENGs. These advancements not only enhance the dielectric properties and surface charge density of TENGs but also improve their environmental sustainability. By optimizing the mixture of cellulose with materials such as conductive nanoparticles and biopolymers, researchers are able to exploit the inherent flexibility and biodegradability of cellulose, thus enhancing the mechanical and electrical characteristics of the composites. This strategic approach not only bolsters the efficiency and longevity of TENGs but also broadens their application spectrum, particularly in wearable electronics and other sustainable energy solutions.

The versatility of cellulose-based TENGs has paved the way for their deployment across a diverse array of applications [31]. In the realm of wearable electronics, these TENGs stand out due to their flexibility and lightweight characteristics, making them ideal for integration into textiles or wearable devices. Here, they play a critical role in powering sensors [46] and various electronic components [52], ensuring seamless functionality. Beyond wearables, cellulose-based TENGs excel in harvesting energy from ambient movements, including human motion, wind, and water waves. This ability allows them to supply power to small electronic devices or sensors situated in remote locations or embedded within wearable technology [56,57,58], thereby facilitating constant operation without the reliance on conventional power sources. Furthermore, in the context of self-powered sensors, these innovative TENGs find applications in environmental monitoring, healthcare, and industrial sensing [31,59,60].

Cellulose-based triboelectric energy harvesting represents a sustainable approach to meet the growing demand for renewable energy sources. By leveraging the natural abundance and favorable properties of cellulose, researchers are developing innovative solutions for efficient energy conversion. Future advancements in material science and device engineering are expected to further enhance the performance and application range of cellulose-based TENGs [28,31], contributing to the global pursuit of green energy technologies.

### 2.2. Lignin-Based Triboelectric Devices

Lignin (Figure 5a), a complex organic polymer found abundantly in the cell walls of plants, particularly in wood and bark [61,62,63], is a promising material for sustainable energy harvesting technologies [64,65,66]. As a byproduct of the paper and pulp industry, lignin is vastly underutilized, often burned for energy or disposed of as waste. However, its inherent properties, such as a high carbon content, biodegradability, and natural abundance, make it an attractive candidate for the development of TENGs [50,67,68,69]. Lignin-based triboelectric devices offer a pathway toward eco-friendly and cost-effective solutions for converting mechanical energy into electrical energy. The operation of lignin-based TENGs relies on the triboelectric effect, a process where materials become electrically charged through friction by coming into contact with and then separating from another material. Lignin, when used as one of the triboelectric layers or as a composite material in TENGs, can effectively generate and transfer electrical charges due to its unique chemical composition and structure. The efficiency of these devices can be further enhanced through chemical modifications or by combining lignin with conductive or other triboelectrically active materials.

Recent advancements in chemical modification techniques have further unlocked lignin’s potential in this domain. These chemical enhancements are designed to introduce functional groups, thereby amplifying the triboelectric effect and, consequently, the efficiency of energy conversion [70,71,72] (Figure 5b–d). The essence of chemically modifying lignin lies in transforming its surface properties to optimize its triboelectric performance. This optimization can be realized through a variety of strategies, including the integration of functional groups that either increase the surface charge density or modify its electron-donating and -withdrawing capabilities. Employing techniques such as esterification [73,74], etherification [75], or the inclusion of conductive nanoparticles or polymers [76,77,78] can significantly modify the electrical and mechanical characteristics of lignin. These alterations render it more conducive for use as a triboelectric material, enhancing its suitability for energy harvesting applications.

Further augmenting the potential of lignin in TENGs is the strategy of surface engineering. This approach seeks to leverage lignin’s intrinsic properties by focusing on its ability to generate and transfer charges upon contact and separation with another material. Key to this strategy is the enhancement of the micro- and nano-scale roughness of lignin surfaces [79,80,81], which can lead to an increased contact area and, thus, improved charge generation. Additionally, increasing the hydrophobicity of lignin can play a crucial role in sustaining its triboelectric performance, especially under humid conditions [82]. Together, these advancements in chemical modification and surface engineering underscore the evolving role of lignin in the field of triboelectric energy harvesting. By capitalizing on lignin’s natural abundance and tailoring its properties to suit energy conversion processes, these techniques open new pathways for the development of efficient and sustainable energy harvesting solutions.

Building upon the recent advancements in the chemical modification and surface engineering of lignin for TENGs, it is evident that lignin’s role in the realm of sustainable energy solutions is becoming increasingly significant. The strategic introduction of functional groups and the enhancement of surface characteristics not only optimize lignin’s triboelectric performance but also enhance its applicability in practical energy harvesting scenarios. The deliberate modification of lignin through esterification, etherification, and the incorporation of conductive materials underscores a tailored approach to enhancing its electrical and mechanical properties, thereby rendering it an effective material for triboelectric applications.

On the other hand, the use of lignin composites as triboelectric materials represents an innovative approach in TENG development, leveraging the natural abundance and unique properties of lignin. Lignin composites are formulated by incorporating lignin with conductive nanoparticles [77,83,84], biopolymers [80], and other natural or synthetic materials [67] (Figure 6). This combination is strategically designed to address and enhance the dielectric properties of lignin that are crucial for triboelectric applications. For instance, the addition of conductive nanoparticles like silver or graphene [85] can significantly improve the electrical conductivity of the composite, while biopolymers can enhance its flexibility and durability. This integration of lignin with various additives and materials not only capitalizes on the inherent characteristics of lignin but also introduces new functionalities through the synergistic effects of the composite materials.

The development of lignin composites for use in TENGs represents a notable innovation, drawing on the intrinsic qualities of lignin as well as its sustainable sourcing. By blending lignin with conductive nanoparticles and biopolymers, these composites are crafted to enhance both dielectric and mechanical properties, key factors in the effectiveness of triboelectric materials. The incorporation of materials like silver or graphene not only boosts the electrical conductivity but also the triboelectric output, making the composites more efficient in energy harvesting.

One of the most compelling aspects of using lignin composites as triboelectric materials is their environmental impact [72,85]. Lignin is a byproduct of the paper and bio-refining industries, and its utilization in TENGs contributes to waste reduction and promotes the use of renewable resources. Furthermore, the biodegradability of lignin and certain additives ensures that these composites align with sustainable development goals, offering an eco-friendly alternative to traditional triboelectric materials.

Lignin-based TENGs harness the unique properties of lignin, such as flexibility and biocompatibility, as mentioned above, to offer versatile applications in energy harvesting and sensor technology. Initially, their integration into wearable electronics capitalizes on the natural movements of the human body, transforming actions like walking into a source of electrical power for devices worn on the body [80,86]. This approach not only aligns with the advancement towards more sustainable wearable technologies but also eliminates the dependency on conventional power sources. Expanding their utility, these TENGs power self-sustained sensors for a variety of monitoring tasks [69,72,86], from environmental observations to health and industrial process control. By generating power from environmental interactions or human motion, they negate the need for external electricity supplies, ensuring continuous operation and reducing maintenance. In essence, lignin-based TENGs represent a step forward in sustainable technology, with applications that range from personal wearable devices to remote sensing and power generation, all while utilizing the renewable and eco-friendly nature of lignin.

While these advancements are promising, the practical deployment of lignin-based TENGs faces several challenges, primarily concerning the scalability of the production processes and the consistency in the properties of the sourced lignin. The variability in lignin’s chemical composition, depending on its biomass source and extraction method, could affect the consistency of its performance in TENG applications. Additionally, the long-term stability and durability of lignin under operational conditions remain areas requiring further exploration. Addressing these challenges through continued research and development could pave the way not only for optimizing the performance of lignin-based TENGs but also for their integration into a broader range of applications, such as wearable devices and portable electronics. This pursuit not only promises to expand the scope of renewable energy technologies but also enhances the sustainability of material utilization in energy systems.

### 2.3. Collagen-Based Triboelectric Devices

Collagen, the most abundant protein in the animal kingdom, is primarily found in connective tissues, such as skin, bones, and tendons [87]. Its unique structural properties, biocompatibility, and biodegradability make it an intriguing material for bioengineering applications. Recently, collagen has been explored as a candidate for TENGs, aiming to harness its potential for energy harvesting and sensing applications [88,89,90]. Collagen-based triboelectric devices are a promising avenue for developing environmentally friendly and sustainable energy technologies that are compatible with biological systems. The operation of collagen-based TENGs relies on the triboelectric effect, where the contact and separation between two dissimilar materials induces a transfer of charges, generating an electrical output. Collagen’s natural abundance and its ability to form stable films make it an excellent material for the triboelectric layer. By incorporating collagen as either the active triboelectric layer or a substrate, researchers can exploit its inherent properties to improve the efficiency and biocompatibility of TENGs. The approaches are discussed in detail below.

Chemically crosslinked collagen represents a groundbreaking approach in the development of triboelectric materials [91], tapping into the vast potential of one of the most widely available proteins in the animal kingdom. This process involves creating covalent bonds between collagen fibers through various agents, such as glutaraldehyde, carbodiimide, or genipin, enhancing both the mechanical robustness and electrical characteristics of collagen [92]. Such improvements not only make collagen more resilient under physical stress but also elevate its triboelectric performance by modifying the surface chemistry and morphology. Crosslinking effectively increases surface roughness and introduces functional groups conducive to the triboelectric effect. Crosslinked collagen’s inherent biocompatibility makes it exceptionally well-suited for wearable and implantable TENG applications, ensuring its compatibility with human tissue. This compatibility, combined with increased durability and resistance to enzymatic breakdown, positions crosslinked collagen as an ideal candidate for sustained use in medical devices and environmental energy harvesters. The chemical modifications from crosslinking fine-tune the electron-donating or withdrawing properties of collagen, enhancing its charge generation capabilities [93].

Building on this foundation, the surface modification of collagen further amplifies TENG functionality [94] (Figure 7a). The introduction of functional groups or additional crosslinking agents refines its electron affinity and charge acceptance capacity. Plasma treatment can also adjust collagen’s surface properties [95], modifying the surface wettability or adding functional groups to improve triboelectric performance. Together, chemically crosslinked and surface-modified collagen pave the way for innovative triboelectric materials. This progression towards sustainable, biocompatible, and efficient energy harvesting solutions suggests a promising future for collagen-based TENGs, offering significant advancements in the realm of energy technology.

The advancements in the development of chemically crosslinked collagen for TENGs represent a significant innovation in the field of energy harvesting technology. This approach effectively utilizes the abundant natural resource of collagen, enhancing its mechanical and electrical properties through chemical crosslinking with agents like glutaraldehyde and genipin. These modifications not only increase the resilience of collagen under physical stress but also improve its triboelectric capabilities by altering its surface chemistry and morphology, thus offering a dual benefit. The introduction of increased surface roughness and functional groups through crosslinking optimizes the electron dynamics at the surface, which are essential for maximizing charge generation in TENGs.

While the potential of crosslinked collagen in TENG applications is considerable, the complexity of the crosslinking process and the need for precise control over the chemical modifications pose challenges. These challenges include ensuring uniformity in treatment across different batches of collagen and managing the potential variability in biocompatibility post-modification. Future research should focus on refining these processes and exploring additional surface treatments to further enhance the triboelectric properties. Advancements in this area could lead to the development of highly efficient, sustainable, and biocompatible TENGs, thereby broadening the scope of their application, not only in the medical field but also in other areas requiring flexible and durable energy solutions. This progression is crucial for the advancement of green energy technologies, aligning with global efforts towards sustainable development.

Hybrid collagen composites have emerged as a groundbreaking category of triboelectric materials, capitalizing on the synergistic combination of collagen with a variety of additional substances to enhance its energy harvesting capabilities (Figure 7b). These composites are meticulously engineered by integrating conductive nanomaterials [90,91], piezoelectric crystals [96], and synthetic polymers [97,98] into a collagen matrix. This integration can be accomplished through several techniques, including physicochemical integration [91,99] or electrospinning [93,100], with each method contributing to the development of composites with distinct structural and functional characteristics. At the core of these composites, the collagen matrix ensures biocompatibility, making the materials particularly appealing for applications in wearable and implantable energy harvesters. The hybrid composition of these composites is strategically designed to optimize both their electrical and mechanical properties, significantly boosting energy conversion efficiency. As a pivotal innovation in the realm of triboelectric materials, hybrid collagen composites open a new avenue toward creating biocompatible, sustainable, and highly efficient energy harvesting devices. They adeptly marry the inherent qualities of collagen with the enhanced functionalities of additional materials, positioning themselves as a vital element in advancing green energy solutions. This innovative approach not only underscores the potential of leveraging natural resources in energy technology but also highlights the role of hybrid composites in shaping the future of sustainable energy harvesting.

The strategic formulation of these composites to optimize both their electrical and mechanical properties exemplifies a forward-thinking approach to material science. This not only maximizes energy conversion efficiency but also broadens the scope of biocompatible energy solutions in medical and environmental applications. However, while the potential of hybrid collagen composites is immense, the challenge lies in maintaining the delicate balance between enhancing functional properties and preserving the natural biocompatibility of collagen. Future research should focus on refining the integration techniques to enhance performance without compromising safety and environmental sustainability. Continued innovation in this direction could lead to the widespread adoption of hybrid collagen composites, significantly impacting the future of sustainable and green energy technologies by providing efficient, durable, and eco-friendly energy harvesting solutions.

Collagen-based TENGs, renowned for their biocompatibility, introduce multifaceted applications in sectors necessitating biological system compatibility [90,91]. In the biomedical field, these TENGs find use in powering both implantable and wearable devices, tapping into collagen’s natural affinity with biological tissues [91]. This makes them ideal for medical applications that require direct integration with the human body. Additionally, their integration into wearable electronics allows for the conversion of human motion into electrical energy, thus providing a sustainable power source for sensors and electronic devices worn on the body [101]. Furthermore, collagen-based TENGs play an important role in environmental monitoring [94]. By utilizing biodegradable materials like collagen, these TENGs present an environmentally friendly option for powering sensors that track environmental changes, aligning with the push towards sustainable and green technology solutions.

### 2.4. Gelatin-Based Triboelectric Devices

Gelatin, a biopolymer derived from collagen through partial hydrolysis [102,103] (Figure 8a), holds significant promise in the realm of TENGs [104]. Its biodegradability, biocompatibility, and versatility, coupled with its intrinsic ability to form films and gels, make it an appealing candidate for eco-friendly and sustainable triboelectric devices. These characteristics facilitate the integration of gelatin into various configurations of TENGs, aiming to exploit mechanical energy from the environment and convert it into usable electrical energy. The fundamental principle behind gelatin-based TENGs is the triboelectric effect, a process where materials become electrically charged through contact and subsequent separation. Gelatin can serve as either a triboelectric layer or a substrate that hosts other triboelectric materials. The unique molecular structure of gelatin, which includes a mix of positive and negative charges [105,106], makes it an excellent candidate for enhancing the efficiency of charge generation and transfer in TENGs.

Chemically crosslinked gelatin has emerged as a promising material in the field of TENGs, offering a novel approach to boost both the efficiency and durability of these devices (Figure 8b,c). Through the chemical crosslinking process [109], where covalent bonds are formed between gelatin molecules using agents like glutaraldehyde [110,111], genipin [112], or carbodiimide [108], gelatin’s mechanical properties and chemical stability are significantly enhanced. This enhancement not only bolsters the physical integrity and thermal stability of gelatin but also improves its resistance to solvents and biodegradability. Such advancements in gelatin’s properties are vital for crafting TENGs that are not only durable and efficient but also environmentally friendly. The procedure endows gelatin with improved durability and stability, making it well-suited to withstand a variety of environmental conditions and thereby extending the lifespan of TENG devices. Additionally, the increased tensile strength and elasticity provided by the crosslinking process benefit triboelectric materials, which often face frequent mechanical stress. Given its lineage from collagen [102,103], crosslinked gelatin retains a high degree of biocompatibility, making it an ideal choice for wearable and implantable energy harvesters. Moreover, the extent of the crosslinking can be finetuned, allowing for the customization of gelatin’s mechanical and electrical properties to optimize triboelectric performance.

Building on this foundation, surface-engineered gelatin introduces an advanced strategy to further optimize the energy harvesting capabilities of TENGs [107]. This involves a variety of surface modification techniques aimed at enhancing gelatin’s triboelectric performance at both the molecular and nanoscale levels. Techniques such as lithography or etching are employed to create micro- or nano-scale patterns on the gelatin surface [105,113,114] (Figure 9), significantly increasing the effective contact area and thereby enhancing the triboelectric effect. These surface modifications lead to an improved charge generation and transfer, culminating in higher electrical outputs from TENGs. The capacity to customize the surface characteristics of gelatin paves the way for TENGs that are specifically tailored for a range of applications, from flexible wearable devices to implantable medical technologies [107,113]. By maintaining the biocompatible and biodegradable nature of gelatin, surface-engineered gelatin-based TENGs stand in alignment with the increasing demand for eco-friendly and sustainable energy solutions, marking a significant step forward in the evolution of energy harvesting technology.

The further modification of gelatin through advanced surface engineering techniques, such as lithography or etching, to create micro- or nano-scale texturing, ingeniously maximizes the contact surface area, thereby significantly boosting the triboelectric effect and the efficiency of charge generation. This tailored approach not only enhances the performance of TENGs but also expands their potential applications, particularly in the fields of wearable and implantable medical devices where biocompatibility and flexibility are paramount. However, while the advancements in gelatin-based TENGs are promising, ongoing challenges such as the scalability of the production processes and the long-term environmental impact of the crosslinking agents used need further investigation. Addressing these issues will be crucial for advancing gelatin-based TENGs towards widespread commercialization. Continued research and development in this area could lead to the creation of highly efficient, durable, and environmentally friendly TENGs, fulfilling the growing demand for sustainable energy solutions in a variety of applications.

Gelatin composites mark a transformative step in the evolution of TENGs, uniting the eco-friendly and biodegradable nature of gelatin with a variety of functional additives to forge a new category of energy-harvesting devices [113,115]. These composites integrate conductive nanomaterials [107,115], piezoelectric crystals [116], and synthetic polymers [108,117] into the gelatin matrix, not only preserving the biocompatibility and environmental sustainability of gelatin but also bestowing improved electrical and mechanical properties essential for effective energy harvesting (Figure 10). The process of creating gelatin composites for TENGs involves the deliberate combination of gelatin with these enhancing materials [115]. This includes mixing gelatin with additives in their solution form prior to gelation, ensuring even distribution throughout the composite. Furthermore, the establishment of covalent or ionic bonds between gelatin and these functional additives significantly bolsters the composite’s stability and durability [108,113,117]. Another key method is electrospinning, which produces fibrous mats of gelatin composites [118]. This technique boosts the surface area and, owing to the resultant nanostructured surface, substantially amplifies the triboelectric effect. Incorporating conductive nanoparticles or piezoelectric materials into the gelatin composites [116] leads to a notable increase in charge density and electrical output, enhancing the efficiency of gelatin-based TENGs. Given its origin from a natural biopolymer, gelatin ensures the non-toxicity and environmental friendliness of the composites, rendering them ideal for applications in wearable and implantable devices. Moreover, the mechanical properties of these composites can be meticulously engineered to range from flexible to rigid, catering to a diverse array of application requirements. By adjusting the type and quantity of additives, the electrical and mechanical characteristics of the composites can be finely tuned, allowing for their optimization to meet specific triboelectric needs. Gelatin composites bring forth an innovative advancement in triboelectric materials, combining the inherent advantages of gelatin with the functional benefits of added materials to develop efficient, biocompatible, and environmentally sustainable TENGs. This strategic integration paves the way for the optimization of triboelectric performance, highlighting the potential of gelatin composites in the future landscape of energy harvesting technology.

The utilization of gelatin, a readily available and biodegradable material, as the foundational component of these composites holds promising potential for the future of sustainable energy technologies. It addresses important environmental considerations while offering a versatile substrate that can be customized to meet the specific demands of various applications, ranging from flexible wearables to more structured, implantable devices. However, challenges remain in ensuring that these composites can be produced at a large scale without compromising their quality or functionality. Moreover, a thorough evaluation of the long-term environmental impacts of the additives and the degradation products of these composites is necessary to substantiate their sustainability claims. Looking ahead, it will be imperative to refine manufacturing processes such as electrospinning to maximize the triboelectric effect while preserving the integrity and functionality of the gelatin matrix. Advancements in these areas could lead to the wider commercialization and application of gelatin-based TENGs, potentially transforming the landscape of energy harvesting technology in medical and environmental contexts.

Gelatin-based TENGs stand at the forefront of versatile and sustainable energy solutions, opening up a wide spectrum of applications across various fields due to their biocompatibility and eco-friendly attributes. In the realm of wearable electronics, the inherent biocompatibility of gelatin allows these TENGs to be effortlessly incorporated into devices worn directly on the body [107,113,114,119]. This integration facilitates the harvesting of energy from human motion, which can then be utilized to power health-monitoring sensors, making everyday wearables more autonomous and less reliant on external power sources. By tapping into these abundant natural energies, gelatin-based TENGs offer a green alternative to conventional batteries, mitigating the environmental impact and enhancing the feasibility of deploying technology in remote or inaccessible areas [59]. Moreover, the biocompatibility of gelatin extends the utility of TENGs into the biomedical domain, where they can be developed as implantable or skin-attachable devices. This application presents a significant advancement in medical technology, providing a reliable power source for self-powered medical devices or sensors. The use of gelatin-based TENGs in this context not only underscores their adaptability to sensitive environments but also highlights their potential to revolutionize the way medical devices are powered, contributing to more sustainable and patient-friendly healthcare solutions.

### 2.5. Keratin-Based Triboelectric Devices

Keratin, a fibrous protein found in hair, nails, feathers, hooves, and animal horns [120,121] (Figure 11a), offers unique properties for the development of TENGs. Its inherent robustness, biocompatibility, and natural abundance in waste materials from the textile and agriculture industries make keratin a compelling material for sustainable energy harvesting technologies. Keratin-based TENGs explore the potential of this protein in converting mechanical movements into electrical energy, aiming at eco-friendly and efficient energy solutions [122,123,124]. Keratin-based TENGs operate via the triboelectric effect, wherein two materials generate an electrical charge upon contact and separation. Keratin, with its complex structure of amino acids, can be utilized as one of the triboelectric layers or as a matrix for embedding other triboelectric materials. Its capacity to form stable films and coatings can be particularly advantageous in creating durable and effective triboelectric surfaces.

Chemically treated keratin, a fibrous protein abundantly found in hair, feathers, nails, and horns, marks a groundbreaking advancement in the development of triboelectric materials [125,126,127] (Figure 11b,c). This innovative approach harnesses keratin’s natural properties through chemical treatments to significantly enhance its energy harvesting capabilities, positioning it as a sustainable and bio-inspired solution for TENGs. This includes the introduction of functional groups onto the keratin surface, effectively increasing its surface charge density and electron affinity to amplify its triboelectric properties [124]. Furthermore, chemical agents are utilized to create crosslinks between keratin molecules [128], which enhance the material’s mechanical strength and stability. This modification may also positively affect its electrical properties, thereby improving its triboelectric performance.

**Figure 11 materials-17-01964-f011:**
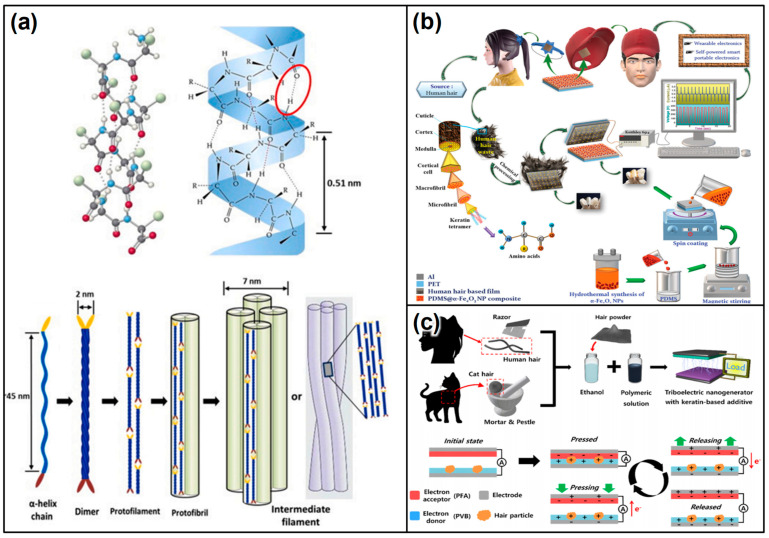
(**a**) Intermediate filament structure of α-keratin: ball-and-stick model of the polypeptide chain, and α-helix showing the locations of the hydrogen bonds (red ellipses) and the 0.51 nm pitch of the helix (reproduced from Ref. [124]. Copyright 2021 Elsevier) Keratin-based triboelectric devices; (**b**) human hair as a recycled material: adding value to the circular bio-economy with waste protein fibers through effective TENG-based mechanical-to-electrical signal conversion technology (reproduced from Ref. [126]. Copyright 2021 Royal Society of Chemistry); (**c**) fabrication process of the keratin-added TENG (reproduced with permission from Ref. [129]. Copyright 2022 American Chemical Society).

Keratin composites represent an innovative advancement in triboelectric materials, ingeniously merging the natural attributes of keratin with a variety of functional additives to forge a novel pathway for energy harvesting. Sourced from abundant natural proteins found in hair, feathers, nails, and horns, keratin provides a biodegradable and biocompatible foundation for these composites. When this foundation is enhanced with functional additives [130] or synthetic/natural polymers [131], the resulting composite exhibits superior electrical and mechanical properties (Figure 12), making it highly suitable for TENG applications. This approach not only taps into keratin’s inherent biocompatibility and biodegradability but also aligns the composites with environmentally sustainable practices, reflecting a commitment to non-toxic and eco-friendly energy solutions. The versatility of keratin composites ensures that the composites can be tailored to meet diverse TENG design specifications. By finetuning the type and quantity of additives incorporated into the keratin matrix, both the electrical and mechanical characteristics of the composites can be optimized to meet the exact requirements of various triboelectric applications. This level of customization underscores the potential of keratin composites to revolutionize the field of energy harvesting, presenting a sustainable, efficient, and adaptable solution for capturing and converting mechanical energy into electrical power.

The transformative potential of keratin-based composites in TENG applications is particularly noteworthy. These composites effectively marry the biocompatibility and eco-friendly aspects of keratin with the robustness and enhanced functionality provided by various additives. This integration aligns perfectly with the current environmental sustainability goals, offering a green alternative to conventional energy harvesting materials. However, the scalability of sourcing and processing keratin at an industrial scale presents a challenge that must be addressed to realize the full potential of these materials. Additionally, the long-term environmental impact of the chemical treatments used to modify keratin should be thoroughly evaluated to ensure that these innovations do not inadvertently contribute to ecological harm. Advancing these technologies will require ongoing research focused on optimizing the properties of keratin composites and expanding their application in practical, real-world settings. 

Keratin-based TENGs present versatile applications, capitalizing on their biocompatibility and sustainable origins to offer innovative energy solutions across various sectors. Firstly, their integration into wearable technologies showcases a significant application prospect. Thanks to their biocompatibility, keratin-based TENGs seamlessly blend with textiles or attach directly onto the skin, efficiently harvesting energy from human movements [125,129]. This ability not only makes personal electronics more sustainable but also enhances the user experience by eliminating the frequent need for external charging. Expanding their utility beyond wearables, these TENGs find critical application in powering self-powered sensors. Given keratin’s natural origin, devices built based on this principle are ideally suited for environmental monitoring and healthcare diagnostics [130,131]. In these contexts, keratin-based TENGs can harness ambient mechanical energies to continuously power sensors that track everything from air quality to patient health indicators or animal movements, providing vital data without the environmental impact associated with traditional power sources. Furthermore, keratin-based TENGs embody a multi-faceted approach to energy harvesting, offering solutions that are not only innovative and efficient but also environmentally responsible and sustainable. Whether integrated into wearable devices, utilized in self-powered sensors, or employed as a means to promote circular economy principles, keratin-based TENGs are poised to make a significant impact across a variety of applications.

### 2.6. Chitin/Chitosan-Based Triboelectric Devices

Chitin, a natural polysaccharide found in the exoskeletons of crustaceans and insects, and the cell walls of fungi, as well as its deacetylated derivative, chitosan [132,133] (Figure 13a), hold significant promise in the field of TENGs [134]. Their biocompatibility, biodegradability, and natural abundance make them attractive materials for developing sustainable and eco-friendly energy harvesting devices. Leveraging the intrinsic properties of chitin and chitosan, researchers are exploring their potential in converting mechanical energy into electrical energy, paving the way for innovative applications in wearable devices, biomedicine, and environmental monitoring. The fundamental principle behind chitin/chitosan-based TENGs is the triboelectric effect, where two materials become electrically charged through contact and subsequent separation. Chitin and chitosan can serve as either the active triboelectric layer or a substrate for other triboelectric materials. Their unique chemical structures contribute to their ability to generate and transfer charges efficiently, making them suitable for various triboelectric configurations.

Chitin and chitosan stand at the forefront of TENG development, offering a blend of biocompatibility, biodegradability, and abundant availability. These characteristics position them as prime candidates for creating eco-friendly and sustainable triboelectric materials, which are essential for the next generation of energy harvesting technologies. The chemical modification of chitin and chitosan plays a pivotal role in enhancing their triboelectric performance [135,136] (Figure 13b,c). This is achieved through several key strategies, such as acylation, grafting, and crosslinking. Acylation serves to introduce acyl groups into the polymers, thereby increasing their hydrophobicity [137]. This modification is crucial for triboelectric applications as it aids in the retention of charges, making the materials more efficient in energy harvesting tasks [138]. Grafting involves the attachment of functional polymers onto the chitin or chitosan backbone [139,140]. This process endows the biopolymers with an enhanced ability to generate and transfer electrical charges, significantly boosting their triboelectric performance (Figure 13d). Crosslinking strengthens the mechanical properties and stability of chitin and chitosan [141,142]. By forming crosslinks between polymer chains, these materials gain durability, making them more suitable for the physical demands of TENG applications [135].

**Figure 13 materials-17-01964-f013:**
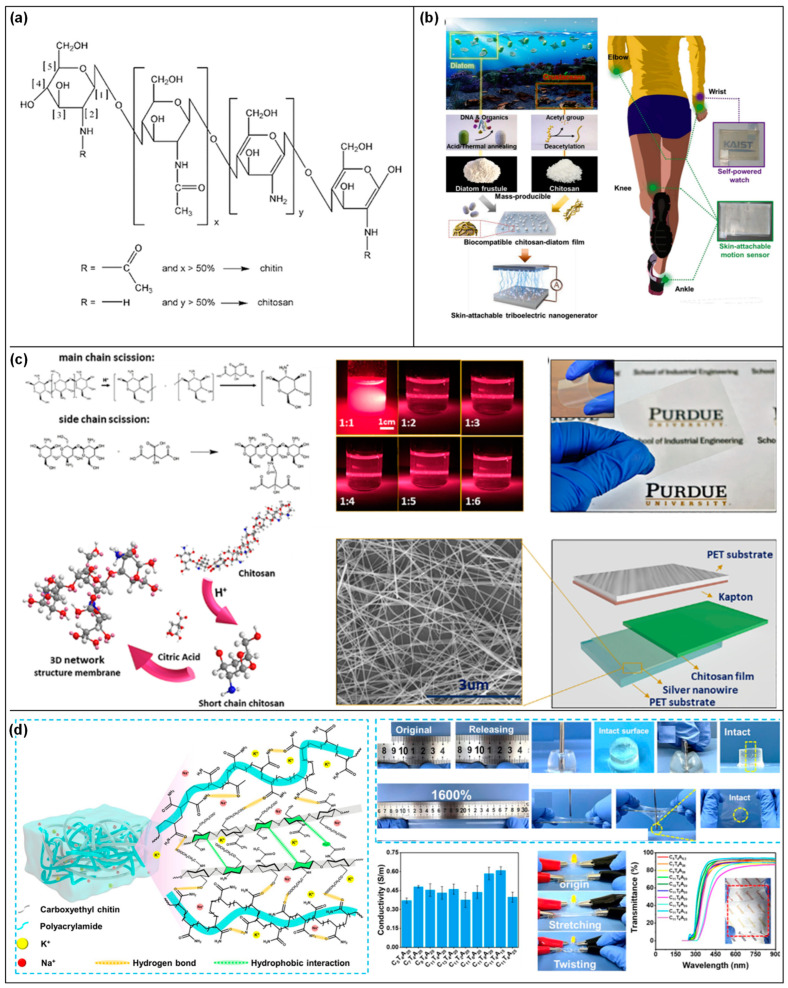
Chitin/chitosan-based triboelectric devices. (**a**) Structure of chitin and chitosan (reproduced with permission from Ref. [133]. Copyright 2015 Wiley); (**b**) skin-attachable and biofriendly chitosan–diatom TENG (reproduced with permission from Ref. [136]. Copyright 2020 Elsevier); (**c**) chitosan–citric acid membrane characterization and triboelectric device (reproduced from Ref. [135]. Copyright 2019 Wiley); (**d**) schematic illustration of molecular interactions among CECT, PAAm in the CTA hydrogel, and its intrinsic properties (reproduced from Ref. [140]. Copyright 2022 Springer Nature).

To further optimize the triboelectric abilities of chitin and chitosan, surface engineering techniques are employed [143]. Surface patterning creates micro- or nano-scale textures on the surfaces of these biopolymers. This patterning increases the effective contact area, which is crucial for enhancing the triboelectric effect by promoting more effective charge generation and transfer during the contact and separation cycles [144]. This treatment ensures the TENGs maintain their efficiency across various conditions.

Through these strategic chemical modifications and surface engineering techniques, chitin and chitosan are transformed into highly effective triboelectric materials. Their natural origin, combined with tailored enhancements, leads to the creation of TENGs that are not only environmentally friendly but also highly efficient in energy harvesting. This innovative approach opens up a wide range of applications, from sustainable power sources to advanced wearable and biomedical devices, marking a significant step forward in the utilization of biopolymers for energy generation and the advancement of green technology.

Chitin/chitosan composites have emerged as a transformative innovation in the domain of TENGs, utilizing the synergistic potential of natural biopolymers combined with advanced functional additives [136,145]. This strategic amalgamation harnesses the biodegradable and biocompatible nature of chitin or chitosan, alongside conductive nanoparticles, piezoelectric crystals, and synthetic polymers, to significantly enhance the composites’ electrical and mechanical properties. Such enhancements are pivotal for the effective transformation of mechanical energy into electrical energy, marking a significant leap in TENG technology. The development of chitin/chitosan composites is carefully orchestrated to amplify their native triboelectric capabilities. By integrating conductive nanomaterials like silver or graphene [146,147], the electrical conductivity of the composite is markedly improved. This improvement facilitates increased charge accumulation and more efficient charge transfer, which are essential aspects of high-performance TENG operation (Figure 14a,b). Furthermore, the incorporation of piezoelectric materials into the composite matrix enriches its capacity to directly convert mechanical stress into electrical energy, thus boosting the overall triboelectric efficiency [148,149] (Figure 14c). Moreover, the fusion of chitin or chitosan with synthetic/natural polymers plays a crucial role in modulating the composite’s mechanical properties [143,150] (Figure 14d). This strategic combination not only enhances the durability and flexibility of the composite but also makes it versatile enough to accommodate a wide range of TENG applications. Considering these characteristics of the composite, chitin/chitosan composites represent a major advancement in TENG technology, showcasing how the integration of natural biopolymers with functional additives can lead to the development of highly efficient, sustainable, and adaptable energy harvesting solutions. Through innovative formulation techniques, these composites are set to play a pivotal role in the evolution of energy harvesting, bridging the gap between sustainability and high-tech energy conversion.

The strategic incorporation of advanced functional additives into the chitin/chitosan matrix is an exemplary model of how interdisciplinary approaches can revolutionize material science, particularly in the context of energy harvesting. While the enhancements in electrical conductivity and mechanical flexibility are promising, it is still challenging to scale these innovations while maintaining their environmental benefits and economic viability. Additionally, the longevity and stability of these composites under real-world operational conditions must be thoroughly assessed to ensure their practical applicability. Moving forward, focusing on optimizing the production processes and enhancing the durability of chitin/chitosan composites will be vital to achieving their widespread adoption in TENG applications. Emphasizing these areas could lead to the creation of next-generation energy harvesters that are not only efficient and versatile but also intrinsically sustainable, marking a pivotal shift in how we approach the design and implementation of energy conversion technologies.

The innovation of chitin/chitosan-based TENGs showcases a remarkable fusion of versatility and environmental compatibility, paving the way for their application across a diverse spectrum of uses. At the forefront of these, their integration into wearable electronics highlights the practical benefits of their biocompatibility and flexibility. Chitin and chitosan are naturally suited for crafting wearable TENGs that efficiently harvest energy from human movements [136,140,143,146]. This ability not only facilitates the operation of wearable health-monitoring devices without the need for conventional power sources but also aligns with the growing trend towards sustainable personal electronics. Expanding their utility, chitin/chitosan-based TENGs excel in powering self-sustained sensors [135,147,151,152]. These innovative devices harness the intrinsic energy-generating potential of chitin and chitosan to autonomously power sensors dedicated to a range of critical functions—from environmental monitoring and agricultural assessments to biomedical diagnostics. This self-powering feature significantly reduces the reliance on external power supplies, offering a green solution to continuous data collection and monitoring in various fields. Furthermore, the potential of these TENGs extends into broader energy harvesting applications. They offer a practical method for powering remote devices, which is especially valuable in locations where traditional power infrastructures are absent or unsustainable. Additionally, their contribution to the energy grid exemplifies a step towards greener energy practices, showcasing the potential of leveraging natural and renewable resources for energy generation.

## 3. Conclusions

The exploration and implementation of biomass-derived materials as triboelectric layers represent a significant stride towards sustainable energy harvesting technologies, as summarized in Table 1. By leveraging the inherent properties of natural biopolymers, enhanced with conductive nanoparticles and piezoelectric crystals through advanced chemical modifications and surface engineering, these materials can offer a promising pathway for efficient and eco-friendly TENGs. Recently, biomass-derived composites have attracted attention due to their superior mechanical and electrical properties. These composites integrate naturally sourced materials such as cellulose and lignin with advanced functional additives, resulting in substantial improvements in mechanical strength and triboelectric efficiency. Advances in the formulation and processing of these composites have led to enhanced charge retention and increased durability, making the devices more effective for long-term energy harvesting applications. Future research is expected to continue refining the composition and manufacturing processes of these biomass composites to optimize their efficiency and sustainability. As these technologies evolve, they are poised to make a significant impact on the renewable energy sector, offering environmentally friendly solutions that can be seamlessly integrated into various applications, from wearable technologies to large-scale industrial uses. This progression underscores the critical role of biomass-derived materials in advancing the next generation of energy harvesting technologies. This innovative approach not only capitalizes on the biodegradability and biocompatibility of biomass-derived materials but also opens new avenues for their application in wearable electronics, self-powered sensors, and renewable energy harvesting, underscoring the potential of biomass in advancing green energy solutions.

## 4. Challenges and Future Perspectives

Despite the substantial advancements in the development of biomass-derived materials for TENGs, certain deficiencies still persist at the current stage. One significant limitation is the inconsistency in the quality and performance of biomass-derived materials, which can vary widely based on the source material and processing methods. This variability can impact the reproducibility and scalability of TENG devices, posing challenges for commercial and industrial applications. Additionally, while enhancements in electrical and mechanical properties have been achieved through the incorporation of high-k materials and conductive additives, the long-term stability and environmental durability of these modified composites under operational conditions still need further exploration.

To address these challenges and advance the field, future research should focus on standardizing production processes to ensure the consistent quality of biomass-derived materials. Moreover, exploring novel bio-derived additives and cross-linking agents could provide new ways to enhance the stability and performance of TENGs. Moreover, there is a significant opportunity to expand the application spectrum of these materials into areas such as bioelectronics and smart textiles, where their environmental benefits and mechanical flexibility could be uniquely advantageous. Emphasizing the development of fully biodegradable triboelectric systems could also align with global sustainability goals, offering a pathway to energy harvesting technologies that are not only effective but also environmentally conscious. These directions not only aim to overcome current limitations but also lead to possible novel applications for future innovations in green energy technologies.

(1) While cellulose-based TENGs are promising for sustainable energy harvesting, challenges in their durability, efficiency, and scalability need addressing. Ensuring long-term stability under various conditions is vital for real-world applications. Efforts to boost energy conversion efficiency through innovations in the material and device are ongoing. Moreover, developing cost-effective, scalable production methods is crucial for commercial viability. Overcoming these obstacles is essential for the widespread adoption of cellulose-based TENGs in green energy solutions.

(2) While lignin-based triboelectric devices show great potential, challenges in material optimization, device durability, and scalability need addressing. Improving lignin’s electrical and mechanical properties is essential for better triboelectric performance. Ensuring long-term stability and durability is crucial for practical applications. Additionally, developing cost-effective and scalable production methods is vital for commercial success. Overcoming these hurdles is key to unlocking the full potential of lignin-based TENGs in sustainable energy solutions.

(3) Collagen-based triboelectric devices, despite their promising potential for sustainable and biocompatible energy harvesting, face challenges in material processing, performance optimization, and biomedical integration. Efficient, scalable processing methods that retain collagen’s properties and biocompatibility are essential. Moreover, enhancing collagen’s triboelectric performance, including the charge density and energy conversion efficiency, requires further research. Additionally, ensuring the long-term stability and biocompatibility of these devices within living organisms is crucial for their successful application in biomedical devices, highlighting the need for comprehensive studies and innovative solutions in these areas.

(4) The potential of gelatin-based triboelectric devices is tempered by challenges related to their hygroscopic nature, material degradation, and the need for optimization and scalability. Gelatin’s moisture sensitivity necessitates solutions for stable performance across humidity conditions. Addressing its long-term stability, particularly for biodegradable uses, is essential. Moreover, scalable fabrication techniques that maintain triboelectric performance while optimizing physical properties are crucial for broadening commercial and practical applications, highlighting the need for innovative approaches in material science and engineering.

(5) Keratin-based triboelectric devices, while promising for sustainable energy harvesting, face challenges in processing and durability, performance optimization, and commercial viability. Efficiently converting keratin into usable forms for TENGs without losing triboelectric properties and ensuring durability is crucial. Research is needed to enhance their electrical and mechanical performance, possibly through advanced composites and architectures. Additionally, scaling production in a cost-effective manner is essential for commercial success, highlighting the need for innovative fabrication techniques to overcome these hurdles.

(6) Chitin/chitosan-based triboelectric devices hold great potential but face challenges in material processing, durability, and optimization for practical applications. Efficient, scalable processing methods are needed to produce high-quality materials for TENGs. Improving the long-term stability and durability of these biopolymers under various environmental conditions is crucial for consistent device performance. Additionally, further research to optimize their electrical and mechanical properties and effective integration into practical applications is necessary, highlighting the need for innovative solutions to leverage their full potential in sustainable energy harvesting.

## Figures and Tables

**Figure 1 materials-17-01964-f001:**
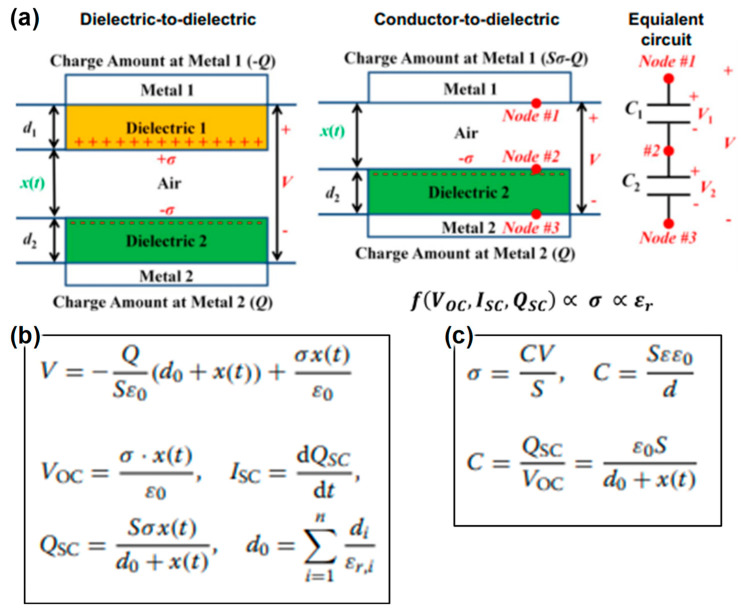
(**a**) Dielectric based triboelectric device and dielectric polarization: theoretical models for parallel plate contact modes and an equivalent circuit diagram for dielectric to dielectric and conductor to dielectric TENG. Triboelectric output performances expressed by (**b**) open-circuit voltage (V_OC_), short-circuit current (I_SC_), and transferred charge (Q_SC_), and (**c**) surface charge density (σ) and capacitance (C), respectively. (Reproduced with permission from Ref. [19]. Copyright 2021 Springer Nature.)

**Figure 2 materials-17-01964-f002:**
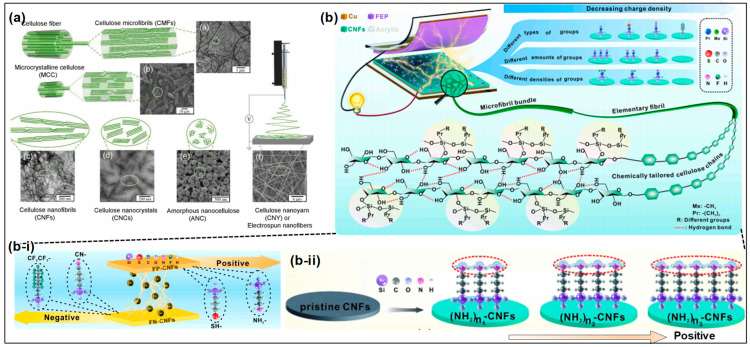
(**a**) Schematic of hierarchical structure of vegetable cellulose nanomaterials (reproduced with permission from Ref. [33]. Copyright 2021 Elsevier); (**b**) schematic diagram of chemical functional groups tailored for use in cellulose nanofibrils to manipulate the charge density; (**b-i**) it contact electrification performance of CNFs with chemically tailored molecular surface modification; (**b-ii**) molecular models with different amounts of -NH_2_ present (n_1_  <  n_2_  <  n_3_) (reproduced with permission from Ref. [36]. Copyright 2021 Elsevier).

**Figure 3 materials-17-01964-f003:**
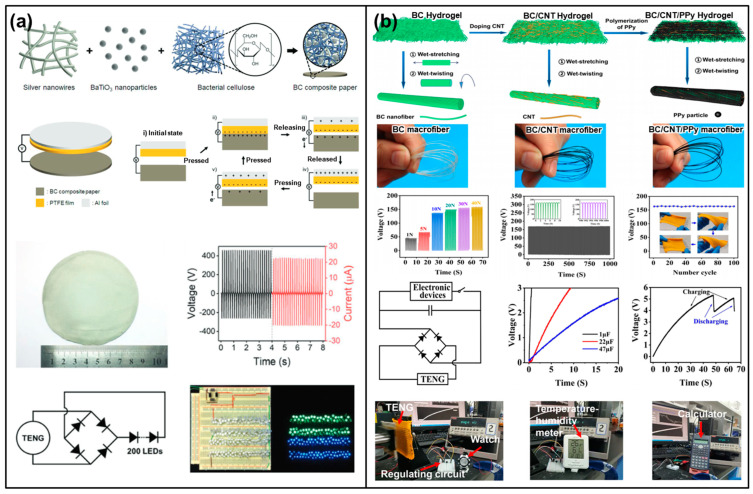
Cellulose composites as a triboelectric material. (**a**) Schematic illustration of the fabrication of highly conductive ferroelectric BC composite paper and the operation mechanism for the BC-TENG in a vertical contact-separation mode. (Reproduced with permission from Ref. [45]. Copyright 2019 Wiley.) (**b**) Schematic illustration of BC, BC/CNT, and BC/CNT/PPy macrofibers’ fabrication and their triboelectric output performances. (Reproduced from Ref. [48]. Copyright 2022 Springer Nature.)

**Figure 4 materials-17-01964-f004:**
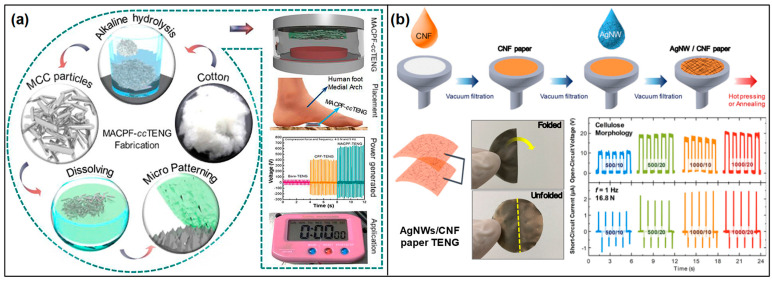
Cellulose composites as a triboelectric substrate. (**a**) Schematic diagram to illustrate the fabrication of CPF-pTENG and triboelectric performance of CPF-pTENG (reproduced with permission from Ref. [51]. Copyright 2019 Elsevier). (**b**) Schematic of AgNWs/CNF paper fabricated using a vacuum-filtration technique and their triboelectric output performance (reproduced with permission from Ref. [52]. Copyright 2018 Elsevier).

**Figure 5 materials-17-01964-f005:**
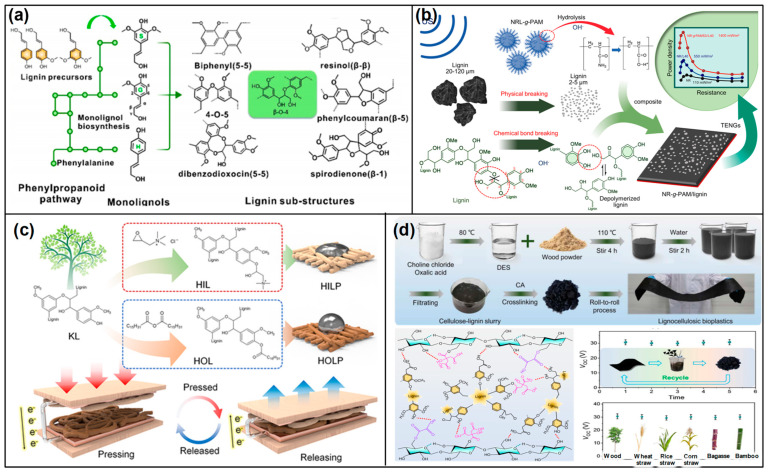
(**a**) Biosynthesis and depolymerization technology of lignin materials (reproduced with permission from Ref. [50]. Copyright 2021 Elsevier). (**b**) Schematic demonstration of the ultrasound-induced transformation of lignin to a triboelectric-active material for NR-based TENGs (reproduced from Ref. [70]. Copyright 2023 American Chemical Society). (**c**) Schematic of the process employed to transform KL into HIL and HOL and the operating mechanism of LP-TENG as an energy harvester (reproduced from Ref. [71]. Copyright 2023 Wiley). (**d**) The fabrication process of the lignocellulose bioplastic through in situ lignin regeneration, crosslinking modification, and the roll-to-roll process and their triboelectric output voltages (reproduced with permission from Ref. [72]. Copyright 2023 Royal Society of Chemistry).

**Figure 6 materials-17-01964-f006:**
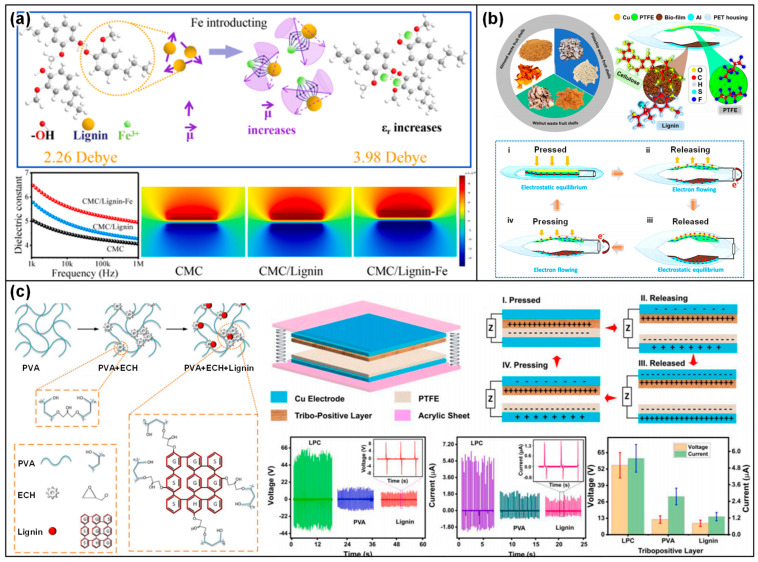
Lignin composite-based triboelectric devices. (**a**) Dielectric property of Fe-doped lignin/CNC composite materials (reproduced with permission from Ref. [83]. Copyright 2023 Elsevier). (**b**) Lignin composite with waste fruit shell (WFS) and the triboelectric device consisting of WFS-based composites. (reproduced with permission from Ref. [67]. Copyright 2022 Elsevier) (**c**) Scheme of the integration of lignin into PVA for the synthesis of composite films in presence of EPCH and their triboelectric output performances. (reproduced with permis-sion from Ref. [80]. Copyright 2024 Elsevier).

**Figure 7 materials-17-01964-f007:**
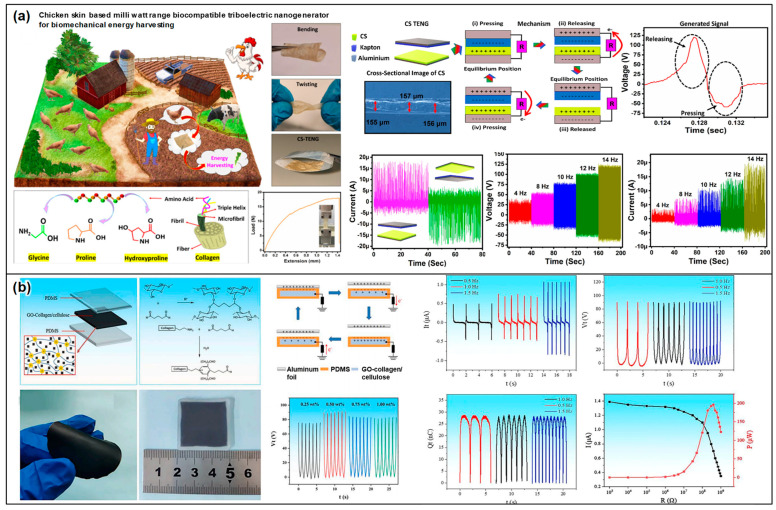
Engineered collagen as a triboelectric material. (**a**) Schematic describing the processing of CS waste for energy harvesting (reproduced from Ref. [94]. Copyright 2023 Springer Nature). (**b**) Schematic diagram of the preparation of GO-CC-TENG and its triboelectric output voltage (reproduced with permission from Ref. [90]. Copyright 2022 Elsevier).

**Figure 8 materials-17-01964-f008:**
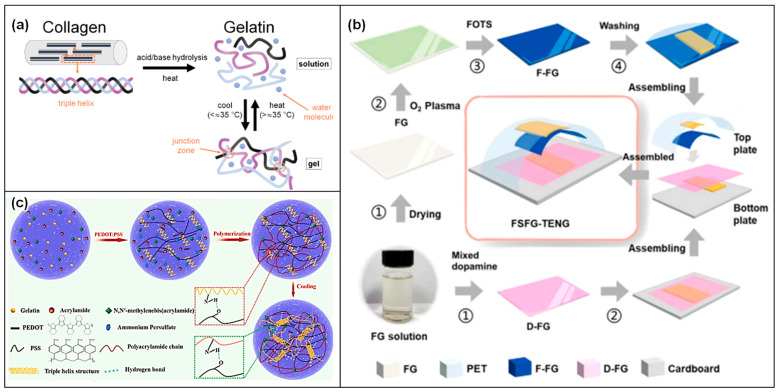
(**a**) Scheme of gelatin extracted from collagen. Crosslinked gelatin used as triboelectric materials; (**b**) schematic illustration of the fabrication process for FSFG-TENG (reproduced with permission from Ref. [107]. Copyright 2021 Elsevier); (**c**) schematic illustration of synthetic procedures of MGP CHs (reproduced with permission from Ref. [108]. Copyright 2020 Elsevier).

**Figure 9 materials-17-01964-f009:**
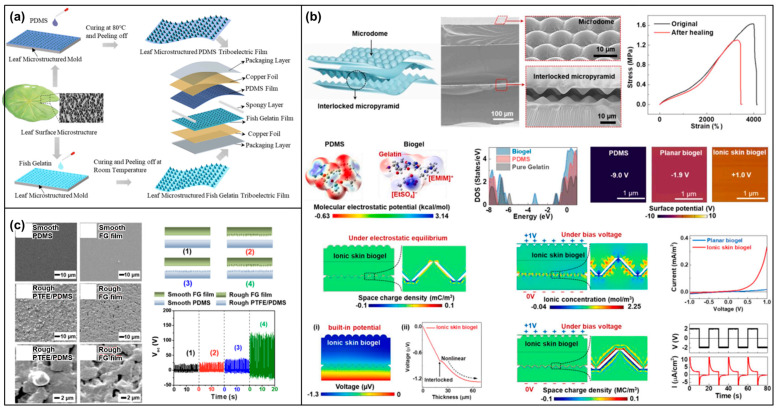
Surface-engineered gelatins for triboelectric applications. (**a**) Schematic illustration of the fabrication processes of the LMFG-TENG (reproduced with permission from Ref. [105]. Copyright 2023 Elsevier). (**b**) Structural and electrical properties of gelatin-ionic skin biogel (reproduced with permission from Ref. [113]. Copyright 2022 Elsevier). (**c**) Output performance of FG-TENGs with different surface roughness (reproduced with permission from Ref. [114]. Copyright 2022 American Chemical Society).

**Figure 10 materials-17-01964-f010:**
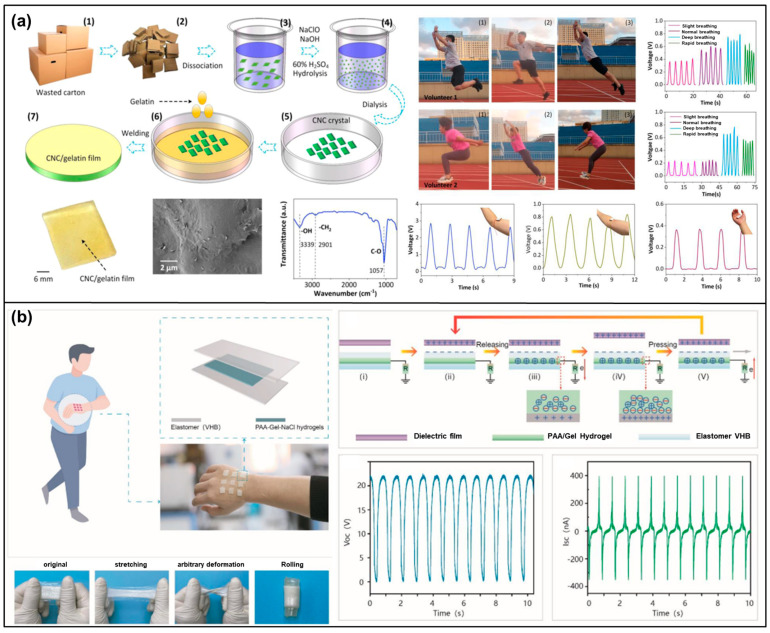
Gelatin composites for triboelectric applications. (**a**) The preparation process of CNC from a wasted box and monitoring of the respiratory status and arm motions (reproduced from Ref. [116]. Copyright 2024 AIP Publishing). (**b**) Schematic diagram of electronic-skin-wearing model based on SH-TENG and the triboelectric output performance (reproduced from Ref. [117]. Copyright 2021 Multidisciplinary Digital Publishing Institute).

**Figure 12 materials-17-01964-f012:**
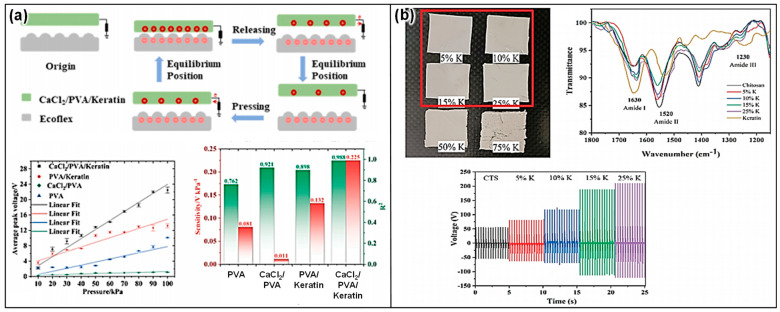
Keratin composites for triboelectric devices. (**a**) Triboelectric working mechanism and output performances of S-TENG (reproduced with permission from Ref. [130]. Copyright 2023 Elsevier); (**b**) synthesized aerogels with different chitosan and keratin contents and their triboelectric output voltage (reproduced with permission from Ref. [131]. Copyright 2024 Wiley).

**Figure 14 materials-17-01964-f014:**
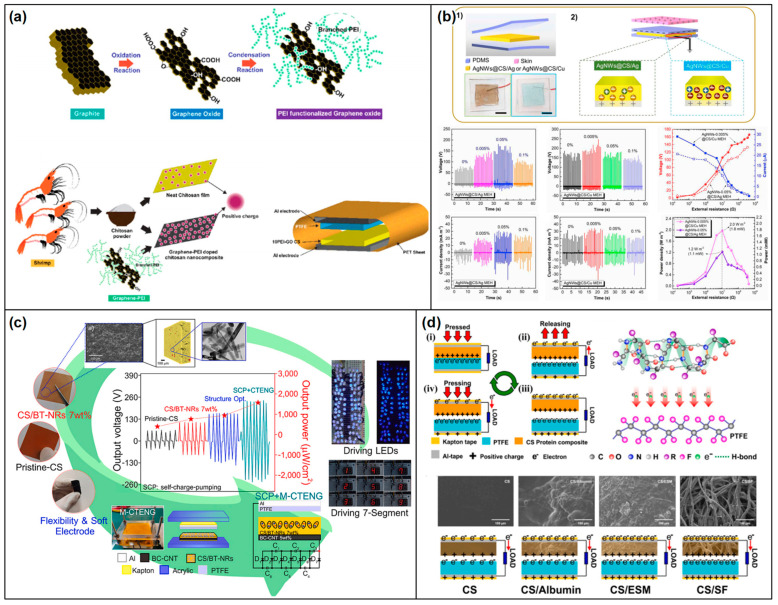
Chitin/chitosan composites for triboelectric devices. (**a**) Schematic diagram of synthesis of GO and the surface functionalization process of GO NPs, PEI-GO CS nanocomposite preparation from the exoskeleton of shrimp, and the TENG device (reproduced from Ref. [146]. Copyright 2024 American Chemical Society). (**b**) Schematic design, configuration, and electrical output of the AgNWs@CS/Ag and AgNWs@CS/Cu hydrogel-based motion energy harvesters (reproduced from Ref. [147]. Copyright 2019 Elsevier). (**c**) Triboelectric–piezoelectric hybrid nanogenerator based on BaTiO_3_-nanorods/chitosan (reproduced with permission from Ref. [148]. Copyright 2021 Elsevier). (**d**) Explanatory diagram of electron transportation based on the molecular structure of protein and PTFE, with a working mechanism in TENG (reproduced with permission from Ref. [150]. Copyright 2021 Elsevier).

**Table 1 materials-17-01964-t001:** Summary of mechanical and electrical performances of biomass-derived triboelectric devices.

Biomass for Triboelectric Devices (Published Year)	Contact Pair Materials	Mechanical Property	Triboelectric Mode	Triboelectric Outputs	Cycle Test	Degradation Test	Applications	Ref.
Cellulose composites (2017–2023)	Bacterial nanocellulose Copper	-	Contact-separation	13 V under 16.8 N and 1 Hz 8.1 μC/m^2^ 4.8 mW/m^2^ at 1 MΩ	-	-	-	[35]
	Amino-CNF FEP	-	Contact-separation	155 V, 17.5 μA, 65 nC 0.22 W/m^2^	10,000 cycles	-	-	[38]
	Allicin-modified CNF PVDF	-	Contact-separation	7.9 V, 1.28 µA/cm^2^, 11.53 μC/m^2^ 18.2 µW/cm^2^ at 10 MΩ	7000 cycles 90 days	-	LED lighting	[39]
	Nitro-CNF Methyl-CNF	-	Contact-separation	8 V, 9 µA	50,000 cycles	-	-	[40]
	CNF-PEI-Ag FEP	-	Contact-separation and relative sliding	100 V, 1.1 µA, 30 nC under 10 N and 2 Hz 0.43 W/m^2^ at 50 MΩ	10,000 cycles	-	LED lighting Touch sensor	[41]
	BC/BTO NP/Ag composite PTFE	-	Contact-separation	460 V, 23 µA 180 µW/cm^2^ at 50 MΩ	10,000 cycles	-	LED lighting	[45]
	BC/CNT/PPy macrofiber PDMS	Tensile strength 449 MPa Young’s modulus 52.2 GPa	Contact-separation	170 V, 0.115 µA/cm^2^, and 0.92 nC/cm^2^ at 4 Hz 54.14 mW/m^2^ at 75 MΩ	1000 cycles	within 108 h	Monitoring human motions	[48]
	5 vol.% Ti_2_NbO_7_ NS/BC composite Al electrode	-	Contact-separation	~36 V and ~8.8 μA 28 µW at 5 MΩ	5000 cycles	-	LED lighting	[43]
	Ethyl cellulose Teflon	Young’s modulus ~84 Gpa	Contact-separation	60 V, 0.12 µA/cm^2^ Instant power 6.8 μW/cm^2^ at 5 Hz RMS power 2.5 μW/cm^2^ at 5 Hz	-	-	-	[44]
	PDA/CNF composite FEP	-	Contact-separation	~205 V, ~20 µA under 60 kPa, 2 Hz ~48.75 μW/cm^2^ at 10 MΩ	10,000 cycles	-	LED lighting Monitoring human motion	[46]
	HECM/PAA/PPy composite hydrogel	G′ (6887.2 Pa) and G″ (1091.1 Pa) Toughness 149.8 KJ/m^3^ Stretchability 316.2% Compressive strength 1.19 Mpa at 78.7% strain	Single-electrode	7.12 V at 1 Hz	>1000 cycles		Strain sensor Powering electronic devices	[47]
	BC/ZnO (ZBC) nanocomposite	-	Single-electrode	57.6 V and 5.78 μA under 2 N, 5 Hz 42 mW/m^2^ at 5 MΩ	1000 cycles	-	-	[49]
	MACPF composite Al electrode	-	Contact-separation	~600 V, 50 μA under 4–5 N, 5 Hz 84.5 W/m^2^ at 200 MΩ	>10,000 cycles	Water droplet after 30 min	Powering electronic devices Monitoring human motion	[51]
	CNF paper AgNW	-	Contact-separation	21 V, 2.5 µA 693 mW/m^2^ at 10 MΩ	-	sonicator for 30 min	LED lighting Capacitor charging	[52]
	BC membrane BC-CNT-PPY membrane	-	Contact-separation	29 V, 0.6 μA 3 μW at 25 MΩ	10,000 cycles	Sugar solution after 8 h	Powering electronic devices	[55]
Lignin composites (2017–2023)	Lignin-starch composite Kapton	-	Contact-separation	1.04 V/cm^2^, 3.96 nA/cm^2^ 173.5 nW/cm^2^	1800 cycles	-	-	[68]
	Lignin CNF composite Cu electrode	Storage modulus ~8 Gpa under 100 °C	Contact-separation	360 V, 28 µA under 20.8 N, 5 Hz 52 µW/cm^2^ at 8 MΩ	72,000 cycles	-	Capacitor charging	[69]
	NR-g-PAM/lignin	Storage modulus ~9300 MPa	Single-electrode	92 V, 5.21 mA/m^2^ 1411 mW/m^2^	5000 cycles	-	LED lighting	[70]
	lignin/polycaprolactone nanofiber (hydrophilic or hydrophonic lignin)	Young’s modulus Hydrophilic lignin ~15.95 ± 4.32 MPa Hydrophonic lignin, Y ~11.8 ± 1.81 MPa	Contact-separation	95 V under 9 N, 9 Hz, 19.2 nC 157 mW/m^2^ at 20 MΩ	100,000 cycles	-	Monitoring human motion (walking and running)	[71]
	Lignocellulosic bioplastic	Tensile strength of 99 Mpa	Single-electrode	31 V, 0.2 mA, and 8 nC at 3.5 kPa 10 mW/m^2^ at 80 MΩ	100,000 cycles	after 12 days by microorganisms	Smart ward and medical monitoring	[72]
	Al-Cu@W/EG-PAM organohydrogel	tensile strength (31.4 ± 3.8 kPa) Stretchability (~800% elongation) Self-adhesion (~31.4 kPa)	Single-electrode	220 V, 4.5 pA, and 0.07 pC	6000 cycles	-	Monitoring human motion (finger- and wrist-bending)	[76]
	Peanut shell powder PTFE	-	In-plane sliding	171.3 V, 24.8 µA 365 mW/m^2^ at 8 MΩ	3000 cycles	-	Powering electronic devices Metal surface anti-corrosion system	[79]
	Lignin/PVA/EPCH composite film PTFE	-	Contact-separation	65 V, 6.44 μA at 3~5 Hz 135.4 mW/m^2^	9000 cycles	-	Self-powered humidity- and pressure-sensing	[80]
	Peanut shell powder PET	-	Contact-separation	910 V, 104.5 μA, and 12 mW by hand excitation force 390 V, 14 μA, and 1.3 mW at 10 Hz	6000 cycles	-	Self-powered pressure-sensing Powering electronic devices	[81]
	CMC/lignin-Fe PVDF	-	Contact-separation	110 V, 8.97 µA, 15.3 nC at 1 Hz 147.19 mW/m^2^	10,000 cycles	-	-	[83]
	Liginin/PLLA composite sheet by laser-induced graphitization	-	Single-electrode	~1.98 mW/m^2^ at 200 MΩ	-	-	Sensing water and plants	[85]
Collagen composites (2021–2023)	Fish bladder film	-	Single-electrode	106 V, 4.56 mA/m^2^, 25 μC/m^2^ 200 mW/m^2^ at 10 MΩ	850 cycles Stable during 12 months	-	Powering electronic devices	[88]
	Eggshell membrane/ZnO composite PTFE	-	Contact-separation	1.2 μA at 2.3 MΩ	3400 cycles	-	-	[89]
	cellulose/collagen/graphene oxide	-	Single-electrode	91.4 V, 0.75 μA, 28.7 nC 31.36 W/m^2^ at 400 MΩ	-	-	LED lighting	[90]
	PCOBE organohydrogel	fracture strength 2.15 MPa, ductility 880% toughness 7.63 MJ/m^3^ Storage modulus 4.84 kPa	Single-electrode	80 V, 5 μA, 50 nC	-	-	Monitoring human motion (EMG, ECG, walking, running, speaking, grasping)	[91]
	Electrospun collagen/PVA/Ag NWs composite PVDF	tensile strength of 3 MPa 90% strain	Contact-separation	118 V, 3.8 nA, 52 nC 21.06 mW/m^2^ at 1GO	1000 cycles	-	Working electronic calculator	[93]
	Collagen/glycine, proline, and hydroxyproline composite film Kapton	Holding 17.5 N load under stress condition (1.4 mm)	Contact-separation	123 V, 20 µA 0.2 mW/cm^2^ at 20 MΩ	>52,000 cycles	-	Powering electronic devices	[94]
Gelatin composites (2020–2024)	Cellulose acetate nanofiber Gelatin/ImClO_4_/Ti_3_C_2_ composite nanofiber	-	Contact-separation	300 V, 10 mA/cm^2^, 18 mC/cm^2^ 500 mW/cm^2^ at 10 MΩ	5000 cycles	within approximately 60 days	-	[106]
	Dopamine-dope fish gelatin FOTS-treated fish gelatin	-	Contact-separation	500 V, 4 μA under 5 N and 5 Hz 100 μW/cm^2^	10,000 cycles	degraded in the soil for 25 days	Human–machine interaction	[107]
	PAM/gelatin/PEDOT: PSS conductive hydrogel	stretchability over 2850% strain)	Single-electrode	383.8 V, 26.9 μA, 92 nC 1250 mW/m^2^ at 30 MΩ	16,000 cycles	-	Monitoring human motions and muscle movements	[108]
	Microdome-structured ionic biogel	E ~104 Mpa stretchability (~4000%), tough (~13,462 J/m^2^)	Single-electrode	11 V, 3 mA/m^2^, 115 µC/m^2^ ~325 mW/m^2^	24,000 cycles	PBS buffer (pH~7.4) solution at 37 °C for 1 h	LCD power Acoustic wave sensing	[113]
	Rough fish gelatin film Rough PTFE/PDMS	-	Contact-separation	130 V, 0.35 μA, 45.8 μW/cm^2^ at 10 MΩ	10,000 cycles	-	Monitoring human body motions	[114]
	CNC/gelatin	-	Single-electrode	248 V, 30 μA, 175 μC/m^2^ 583 μW at 20 MΩ	>30,000 cycles	-	Running and jumping, training monitoring	[116]
	PAA-Gel-NaCl hydrogel	800% (uniaxial strain) Storage modulus 10~20 kPa	Single-electrode	~6 nC, ~22 V, and ~400 nA ~2.9 µW/cm^2^ at ~140 MΩ	12,000 cycles	-	Touch/pressure-sensing	[117]
	Nanostructured gelatin films Electrospun PVA	-	Contact-separation	900 V, 10.6 mA/m^2^ under 50 N, 5 Hz 5 W/m^2^ at 80 MΩ	15,000 cycles	Natural water completely in about 40 days	-	[118]
	Microstructured gelatin PDMS/Mxene	-	Contact-separation	417.39 V, 12.01 μA 170 μW/cm^2^ at 10 MΩ	10,000 cycles	150 s in water at a constant temperature of 75 °C	Powering electronic devices Forefoot and heel applications	[119]
Keratin composites (2020–2024)	Hair particle-embedded PVB PFA	-	Contact-separation	296 V, 37.6 μA 0.6 mW/cm^2^ at 10 MΩ	9000 cycles	-	Powering electronic devices	[129]
	CaCl2/PVA/Keratin	Strain (~300%) Stretchability (200%)	Single-electrode	24 V, 0.225 V/kPa	18,000 cycles	-	Sensor location layout and wearable data acquisition system	[130]
	Chitosan/ketratin composite PTFE	-	Contact-separation	322 V, 32.2 μA 14.4 W/m^2^ at 0.1 MΩ	8000 cycles		Powering electronic devices	[131]
Chitin/Chitosan composites (2018–2023)	Chemically treated chitosan Kapton	-	Contact-separation	16.2 V, 125 uA/m^2^, 6.8 uC/m^2^	-	-	Data-driven learning for car speed sensing	[135]
	Chitosan–diatom film FEP	-	Contact-separation	150 V 15.7 mW/m^2^ at 100 MΩ	-	-	Self-powered watch and skin-attachable motion sensor	[136]
	Carboxyethyl chitin/polyacrylamide hydrogel	strain (1586%), self-adhesion (113 kPa for pigskin) toughness 1299.71 kJ/m^3^ Elastic modulus 66.62 kPa	Single-electrode	58 V, 52 nC, 1.2 μA 1.17 W/m^2^ at 1.5 GΩ	10,000 cycles	-	Human–machine interface	[140]
	Chitosan with glycol Ecoflex	-	Contact-separation	13.5 V, 42 nA 17.5 µW/m^2^	-	-	-	[143]
	Chitosan/1 wt% Kaolin Silicone	-	Contact-separation	996 V, 26.5 W/m^2^ at 1.1 MΩ	-	-	-	[144]
	Chitosan–glycerol film PTFE	-	Contact-separation	~130 V, 90 μC/m^2^	10,000 cycles	-	Self-powered sweat sensor Self-powered gait phase detector	[145]
	AgNWs@CS/Ag hydrogel AgNWs@CS/Cu hydrogel	-	Single-electrode	175 V, 25.1 mA/m^2^, 1.2 W/m^2^ at 10.5 MΩ 218 V, 34.44 mA/m^2^, 2 W/m^2^ at 10.5 MΩ	Reduced within 24 h (22 °C, RH of 58%)	-	Self-powered temperature–stress dual sensor	[147]
	CS/BT-NRs PTFE	Tensile strength 15.7 Mpa Elongation at break 259.1%	Contact-separation	111.4 V, 21.6 μA 756 μW/cm^2^ at 4 MΩ	3000 cycles	-	Powering electronic devices	[148]
	BT-NPs embedded into CS PTFE	-	Contact-separation	110.8 V and 10 μA 431.8 μW at 6 MΩ	2000 cycles	-	LED lighting	[149]
	CS protein composite PTFE	-	Contact-separation	~77 V and ~13 µA 22.4 μW/cm^2^ at 3 MΩ	-	long-term stability within 9 weeks	LED lighting	[150]
	Chitin-hydrogel film	Tensile strength 84.7 MPa elongation at break 14.5% Young’s modulus 2.4 Gpa	Single-electrode	182.4 V, 4.8 μA 1.25 W/m^2^ at 3 Hz	2000 cycles	after burying in soil for 25 days	Powering electronic devices Monitored real-time pulse signals from carotid artery human–machine interface with non-contact sensation	[151]
	Polyacrylic acid/nanochitin composite hydrogel	400 kPa tensile strength strain range of 0–500%	Single-electrode	71 V, 7.8 μA 1.06 W/m^2^ at 5 MΩ	200 cycles	-	pressure sensor	[152]

## Data Availability

No new data were created.

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
