# Peer review of "Advanced Triboelectric Applications of Biomass-Derived Materials: A Comprehensive Review"

_materials, 2024, doi:10.3390/ma17091964_

Round 1
Reviewer 1 Report
Comments and Suggestions for Authors
The review is an excellent attempt to compile the research conducted on the biomass-derived materials-based triboelectric system. The review is excellently written and covers a broad aspect of the recent research. Moreover, this review is very well-structured and rich in content, and I am very certain that it will attract a lot of interest from the scientific community. However, before accepting this manuscript, the authors should address some minor issues. The comments and suggestions about this work are described as follows:
1. In the Introduction section, even though the authors already included many citations, it is still suggested to cite more papers to enrich their background. For instance, in the first sentence of Section 1.2, there are no citations.
2. All the figures from previous literature are blurry, especially for those English words. It is suggested to ask the original authors for these papers for high-resolution figures.
3. For all the acronyms, after the first definition, the authors don’t need to define them again. For instance, on page 10, the authors defined TENG again. They should double-check all the acronyms.
4. It is great to see that the authors cited many good literature as examples. However, I didn’t see a very detailed discussion on these examples. It is suggested to provide more conclusions of the authors’ own, instead of just copying the conclusion from the literature.
5. Biomass-derived materials-based triboelectric system is a rapidly evolving field, and each passing year presents new developments that are way better than the previous ones. In such a dynamic research area, it is expected that a review must provide the latest progress in the area to justify its relevance to the field. I would suggest the authors update the review with significant reports from the last 2 years (2022 - 2024) to make it more timely and relevant.
6. In the conclusions and perspectives, the author can add more details focusing on describing the deficiency of the current stage for biomass-derived materials-based triboelectric systems and pointing out the potential or suggested direction for novel applications in future work.
Comments on the Quality of English LanguageThe authors have good English skills. However, there are still some oral expressions in the manuscript. The authors should double-check the whole manuscript.
Author Response
We would like to thank referees for their encouraging evaluation and constructive comments on our manuscript. In accordance to comments/suggestions from referees, careful revision has been made. The revised parts were highlighted in red. Please find an attached file containing point-by-point response to the comments from the reviewers.

Reviewer 2 Report
Comments and Suggestions for Authors
The paper prepared by authors Chan Ho Park and Minsoo P. Kim is interesting and well prepared as a general review. The authors provide many examples and explanations but the report lacks details.
I would recommend adding some details and comparing the materials, methods, and performances of the biomaterial-based eco-TENGs. Please compare mechanical, electrical and other properties. The presented materials are biomaterials but not all of them are biodegradable in the natural environment. Biodegradable materials can also cause water pollution with micro or nanoparticles.
Author Response

(The authors gave the same response as above.)

Reviewer 3 Report
Comments and Suggestions for Authors
Revision
In abstract:
Sugestion: In this review, we display an overview of the emerging field of advanced…
For example, without we: This review presents an overview of the emerging field of advanced triboelectric applications that utilize the unique properties of biomass-derived materials.
Additionally, it discusses the challenges and opportunities in using these materials for sustainable and environmentally friendly energy solutions.
In 1.1 introduction
You write: For instance, cellulose nanofibers from wood pulp are used in triboelectric nanogenerators (TENGs) for their lightweight, high surface area, and piezoelectric properties, offering a sustainable alternative to traditional materials.
What is the traditional materials?
In 1.2 section
You write:
This underscores the importance of choosing the right ma terials for the triboelectric pairs and crafting the device structure with precision. Reflecting on their operational fundamentals, four distinct models of triboelectric devices have been identified, all of which utilize dielectric materials in their triboelectric layers [9].
Which is the four distinct models of triboleletric devices? The four fundamental modes of triboelectric nanogenerators. vertical contact separation mode; in-plane contact-sliding mode; (single-electrode mode and freestanding triboelectric-layer mode?
Page 3
You write:
This review also addresses the challenges and opportunities in harnessing biomass-derived materials for triboelectric applications. Challenges such as enhancing the durability and efficiency of these materials in TENGs are critical for their broader adoption. Conversely, the opportunities lie in the untapped potential of various biomass sources and their conversion into high-performance triboelectric materials.
How can enhancing the durability?
Page 5, figure 3
The comparison of current, voltage and frequency parameters is crucial. Graphs that illustrate these values enable you to ascertain whether you possess the capability to power a low-consumption sensor.
Page 6
You write: The versatility of cellulose-based TENGs has paved the way for their deployment across a diverse array of applications [14]. In the realm of wearable electronics, these TENGs stand out due to their flexibility and lightweight characteristics, making them ideal for integration into textiles or wearable devices.
In textiles application, won't the problem of being biodegradable influence their life cycle?
Page 7, figure 5
What is LB? Lignocellulose bioplastic?
Page 8, figure 6 and same for page 11 figure 7
Plotting voltage vs frequency or current vs frequency enhances our understanding of the available power.
Page 19, figure 13
Suggestion: Would it be better to place Figure 13 after the text?
The same for figure 14.
Page 21, conclusion
It would be very helpful to have a table comparing the advantages and disadvantages of each triboelectric generator material, including their power as a function of frequency. Although each material has been described, it is still unclear which is superior.
Comments on the Quality of English LanguageMinor editing of English language required
Author Response

(The authors gave the same response as above.)

Reviewer 4 Report
Comments and Suggestions for Authors
In this paper, the biomass triboelectric nanogenerators are reviewed in detail, and the current development status, challenges and prospects of biomass triboelectric nanogenerators are expounded. The topic is worthy to be reviewed. However, there are some small improvements that need to be made before the paper can be accepted and published. My detailed comments are as follows:
1.The similarities and differences of the materials should be properly analyzed.
2.The layout of the article needs optimization.
3.This paper lacks substantive application for the future prospect of biomass triboelectric nanogenerators.
4.In reference, (44), (59), (68), (70), (76), (83), (85), (89), (93), (100),the reference format is inconsistent with other references.
Author Response

(The authors gave the same response as above.)

Round 2
Reviewer 2 Report
Comments and Suggestions for Authors
I would recommend accepting the paper in the present form.